# Fusion of Hsp70 to GFP Impairs Its Function and Causes Formation of Misfolded Protein Deposits under Mild Stress in Yeast

**DOI:** 10.3390/ijms241612758

**Published:** 2023-08-14

**Authors:** Erika V. Grosfeld, Anastasia Yu. Beizer, Alexander A. Dergalev, Michael O. Agaphonov, Alexander I. Alexandrov

**Affiliations:** 1Bach Institute of Biochemistry, Federal Research Center of Biotechnology of the RAS, 119071 Moscow, Russia; 2Moscow Institute of Physics and Technology, National Research University, 141700 Dolgoprudny, Russia; 3Weizmann Institute of Science, Rehovot 7610001, Israel

**Keywords:** yeast, chaperone, misfolding, deposit, IPOD, Hsp104, oxidative stress

## Abstract

Protein misfolding is a common feature of aging, various diseases and stresses. Recent work has revealed that misfolded proteins can be gathered into specific compartments, which can limit their deleterious effects. Chaperones play a central role in the formation of these misfolded protein deposits and can also be used to mark them. While studying chimeric yeast Hsp70 (Ssa1-GFP), we discovered that this protein was prone to the formation of large insoluble deposits during growth on non-fermentable carbon sources under mild heat stress. This was mitigated by the addition of antioxidants, suggesting that either Ssa1 itself or some other proteins were affected by oxidative damage. The protein deposits colocalized with a number of other chaperones, as well as model misfolded proteins, and could be disassembled by the Hsp104 chaperone. Notably, the wild-type protein, as well as a fusion protein of Ssa1 to the fluorescent protein Dendra2, were much less prone to forming similar foci, indicating that this phenomenon was related to the perturbation of Ssa1 function by fusion to GFP. This was also confirmed by monitoring Hsp104-GFP aggregates in the presence of known Ssa1 point mutants. Our data indicate that impaired Ssa1 function can favor the formation of large misfolded protein deposits under various conditions.

## 1. Introduction

The accumulation of misfolded proteins is typical of many neurodegenerative disorders, including Huntington’s and prion disease, as well as aging and various types of environmental stress. This accumulation is thought to occur due to the failure of cellular protein quality control mechanisms, such as chaperones controlling protein misfolding and aggregation, as well as the ubiquitin proteasome system and autophagic machinery, which degrade proteins. Recent studies have shown that various types of misfolded proteins, be they amyloid or non-amyloid in nature, can, under certain circumstances, gather into spatially defined compartments [1,2,3,4]. Several reports indicate that the formation of these compartments can play a role in mitigating the deleterious consequences of protein misfolding [4,5,6].

Several types of deposition sites have been described, including the IPOD (Insoluble Protein Deposit) and IPOD-like complexes [1,7], which, in yeast, are associated with the vacuole, the JUNQ (INQ) (Juxtanuclear quality control; Intranuclear quality control), which is either juxtanuclear or intranuclear [1,8], and various types of other foci [3,9,10,11,12]. JUNQ foci are associated with proteasomal degradation, as well as proteosomal markers, while IPOD foci seem to be formed either by substrates that cannot be degraded by the proteasome or during impaired proteasomal activity [1].

Different proteins have been demonstrated to form IPOD-like deposits (reviewed in [7]), e.g., VHL (Von Hippel–Lindau tumor suppressor), a protein which is misfolded in the absence of its partner elongin proteins and which forms IPODs in response to the inhibition of proteasome function in yeast [1], amyloids formed by huntingtin, and the yeast prion proteins Sup35 [13,14] and Rnq1 [1], which form IPODs when overexpressed in otherwise non-stressed conditions.

Chaperones, including Ssa1/2 and Hsp104, are known to colocalize with IPOD deposits [1,15] and have been shown to participate in their dissolution [1,13].

In this work, we observed that a C-terminally GFP-tagged chimera of the Hsp70-family protein Ssa1 from the GFP-fusion library [16] forms IPOD-like inclusions under various mild stress conditions. We termed these inclusions HADS (Hsp70-associated deposit site). HADS can be disassembled by Hsp104 and seem to be a consequence of impaired Ssa1 function caused by GFP, since the conditions which promote HADS formation by Ssa1-GFP do not do so for wild-type Ssa1 or a chimera tagged by Dendra2. They are also likely to include oxidatively damaged proteins, because less HADS formed in the presence of antioxidants. Our results provide a convenient model with which to study the formation of IPOD-like protein deposits and to identify conditions which can trigger proteotoxic stress. They also highlight the complex ways in which various fusion proteins can affect the activity of proteins in cells. 

## 2. Results

### 2.1. Ssa1-GFP Forms Large Protein Deposits under Conditions of Growth on a Non-Fermentable Carbon Source and Heat Stress

While working with a strain from the GFP-fusion library [16], harboring Ssa1-GFP, we noticed that this protein diffusely distributed in log-phase cultures growing on YPD medium (rich medium with glucose as a carbon source), but formed various foci, including dramatic single large foci, in cells that were grown to high densities (Figure 1A). Such a condition is characterized by the depletion of glucose, which is converted into ethanol. This suggested that either starvation, ethanol stress or the utilization of non-fermentable carbon sources could be the cause of this relocalization. To check this, we assayed the foci formation of Ssa1-GFP with glycerol as the sole carbon source (YP-Gly) and observed robust foci formation (Figure 1A). 

Growth on non-fermentable carbon sources increases the amount of oxidatively damaged proteins [17]. This suggested to us that other types of proteotoxic stress might also trigger HADS formation, and, indeed, we discovered that growth under heat-stress conditions (37 °C) of log-phase cultures in YPD (with glucose as the sole carbon source) also resulted in foci formation (Figure 1A). 

Since the formation of reactive oxygen species can depend on the activity of mitochondria, we also tested whether HADS formation in conditions of stationary growth phase could occur in the absence of the mitochondrial genome by obtaining rho^0^ mutants. To our surprise, HADS formation did not depend on the presence of the mitochondrial genome, i.e., oxidative phosphorylation was not essential for HADS formation (Figure 1B). This might be due to the fact that the absence of the mitochondrial genome, and the resulting mitochondrial dysfunction, increases the amount of reactive oxidative species in the cell, as evidenced by increased fluorescence of several redox-sensitive dyes, as well as increased amounts of carbonylated proteins in rho^0^ cells [18,19]. Oxidative stress was also shown to depend on RAS signaling, specifically, Yno1, an ER-residing NADPH oxidase [18]. 

### 2.2. HADS Are IPOD-like

Since various types of foci associated with misfolded protein deposition have been described, we decided to determine whether HADS belonged to one of these types. We tested whether Ssa1-GFP foci colocalized with proteasomes (Sem1-RFP), which are usually present in JUNQ [1], known IPOD-markers (Hsp42, Hsp104), as well as the paralog of Ssa1–Ssa2, which is nearly identical to Ssa1. We also tested for several model misfolded proteins which have been shown to be deposited both in the IPOD and in other misfolded protein deposition sites, i.e., Von Hippel–Lindau tumor suppressor (VHL) fused to mCherry and model huntingtin containing a polyproline stretch (Htt-polyPro-Q103-RFP) (Figure 2). These experiments showed that Ssa1GFP foci did not colocalize with proteasomes (which rules out HADS being similar to the JUNQ) and associated partially with VHL (which marks both the IPOD and JUNQ, depending on proteasomal status) and Hsp42 (an IPOD marker). The latter protein was mostly localized at the edge of the HADS. However, HADS efficiently colocalized with Ssa2-RFP and Hsp104-RFP (which is an IPOD marker). The colocalization with Htt-polyPro-Q103-RFP was partial, i.e., in some cells, huntingtin associated with the HADS as a ring on the outer edge of the HADS, while in other cells, Ssa1-GFP and huntingtin formed adjacent foci which contained differing proportions of the two proteins. Overall, these data suggest that HADS are somewhat similar to the IPOD [7]. 

### 2.3. Ssa1-GFP Foci Formation Is Reversible in an Hsp104-Dependent Manner

Since HADS assemble during growth on YP-Gly and do not do so on YPD during logarithmic growth, we tested whether these foci were disassembled after a change of carbon source. This was indeed the case; moreover, using targeted deletion of the *HSP104* gene (Figure 2B), we showed that HADS dissolution was dependent on the activity of this chaperone. 

### 2.4. Ssa1-GFP Foci Formation in Respiration Conditions Is Related to Oxidative Stress

Because HADS form in proteotoxic conditions (heat stress and utilization of a non-fermentable carbon source), which is likely to increase the flux of proteins through the proteasome, and the known involvement of Ssa1 in this flux [20], we tested whether the addition of antioxidants N-acetyl-cytstein (NAC) and ascorbic acid (AA) could mitigate HADS formation, presumably by reducing the amount of oxidatively damaged proteins sent for degradation. The number of cells with HADS was reduced considerably (Figure 3A,B), and thus, we can propose that oxidative damage to either Ssa1-GFP or to other proteins is an important factor in the formation of HADS inclusions under conditions where respiration is active. 

Notably, the addition of antioxidants also reduces the temperature-sensitive phenotype exhibited by Ssa1-GFP bearing cells (see below) (Figure 3C).

### 2.5. Effective HADS Formation Is Likely Due to the Dysfunctional Activity of the Ssa1-GFP Protein

As noted above, Ssa1-GFP foci colocalize with Ssa2-RFP (Figure 2) when these two proteins are co-expressed; however, to our surprise, on their own, neither Ssa2-RFP nor Ssa2-GFP formed HADS during growth on YP-Gly or in YPD at 37 °C, which are conditions conducive to HADS formation for Ssa1-GFP. This suggested that Ssa1-GFP was defective in its function and that this was the reason for foci formation. In agreement with this, a different C-terminal fusion of Ssa1, Ssa1-Dendra2, as used in our previous work, did not form large aggregates when grown on YP-Gly medium or during growth at 37 °C [21]. In order to further verify the dysfunction of Ssa1-GFP, we assayed the growth rate of the strain harboring Ssa1-GFP, as well as those of control strains (Ssa2-GFP, Hsp104-GFP and wild-type) and those lacking Ssa1, at 37 °C (Figure 3C). 

The Ssa1-GFP strain proved to be sensitive to this temperature, and since it has been previously shown that individual deletions of SSA1 and SSA2 do not cause temperature sensitivity but that double deletion of SSA1/SSA2 does [22], Ssa1-GFP should have a negative effect on wild-type Ssa2, which is nearly identical in its amino acid sequence, except for a small region on the C-terminus. This is probably due to the fact that Ssa2 can colocalize with HADS but does not form them in the absence of Ssa1-GFP (Figure 2A).

Finally, in order to determine whether wild-type Ssa1 formed assemblies similar to HADS, we performed centrifugation of lysates obtained from WT cells and Ssa1-GFP expressing cells grown on YP-Gly or YPD medium. We observed considerable amounts of Ssa1-GFP in the pellet fraction obtained from YP-Gly and not YPD, and the amount of protein was sufficient to observe it using Coomassie staining. The identity of the protein band was confirmed by MALDI-TOF. However, we did not observe a similarly abundant protein band corresponding to wtSsa1 in the control strain (Figure 3D). This allowed us to conclude that wild-type Ssa1 does not form comparable amounts of large aggregates that are similar to HADS. We cannot exclude the possibility of it forming smaller aggregates with distinct sedimentation/stability characteristics. 

These data suggest that the formation of large HADS by Ssa1-GFP is likely to be a consequence of Ssa1 dysfunction; notably, Ssa1-Dendra2, which did not form HADS under the conditions in which Ssa1-GFP did, was able to form considerably smaller HADS-like foci in cells that were taken from colonies that had grown for 2 days on solid YPD medium (Figure 3E), but not in conditions where Ssa1-GFP formed HADS effectively. Thus, inclusions similar to HADS may form under physiological stress conditions; however, these need to be characterized in further studies.

In order to independently verify that a dominant-negative perturbation of Ssa1 activity could result in the formation of protein aggregates, we expressed two SSA1 alleles with known dominant-negative point mutations (*ssa1-21* and *ssa1-K69M*, both of which impair prion propagation in yeast) in a strain with GFP-tagged Hsp104. Hsp104-GFP was chosen as a marker for the detection of HADS-like aggregates, since its RFP-tagged counterpart participates in the original HADS formed by Ssa1-GFP (Figure 2A). While the presence of these mutants did not have a strong effect on aggregate formation under conditions of logarithmic growth (Figure 3G), once the cells had reached stationary phase, the proportion of aggregate-bearing cells was significantly increased for both Ssa1 mutants tested (Figure 3F,G). Since the stationary phase is a stress condition that favors HADS formation (Figure 1A), we can assume that the observed aggregates were HADS-like protein deposits that recruited Hsp104 and probably other chaperones. The small size of the aggregates might also explain why we could not detect large amounts of sedimented wild-type Ssa1 in cells grown in YP-Gly (Figure 3D). The small size of the aggregates would reduce the share of the protein involved in the aggregates, as well as, possibly, their stability during and after cell lysis. 

## 3. Discussion

We observed that the fusion of GFP to the Ssa1 protein impairs the function of this protein, and under mild, presumably proteotoxic stress conditions, this impairment results in the formation of large insoluble Ssa1-GFP-containing protein deposits. This is in good agreement with previous data on the formation of insoluble protein deposits in cells lacking Ssa1/Ssa2 [10], as well as on the role of Ssa chaperones in the transport of proteins for degradation by the proteasome [20]. Since proteasome inhibition is often required for the observation of various types of misfolded protein deposition sites [1], it is highly plausible that impairment of the role of Ssa1 in protein degradation is the cause of HADS formation. 

It is not very clear whether the formation of HADS is due to the increased propensity of Ssa1-GFP to aggregate or if the impairment of its function causes the various substrate proteins to gather into the foci. Since we observed noticeable localization of misfolded proteins (VHL and huntingtin) into the HADS, the second option has at least some support. Because we also show that antioxidants reduce the efficiency of HADS formation, it is likely that oxidative damage of Ssa1-GFP or other proteins plays an important role in HADS formation. 

Although the phenomenon we observed seemed not to be directly relevant to the physiology of normal cells, we have demonstrated that a different fusion of Ssa1, which is evidently less impaired in its function (since it does not display a temperature growth defect), does form foci under conditions of growth in colonies, although their relation to HADS was not assessed. This may indicate that the formation of insoluble misfolded protein deposition sites is a physiologically relevant phenomenon; however, the conditions under which it is observed have not been elucidated. Additionally, our data suggest that the C-terminus of Ssa1 may play an important function, and that its impairment causes HADS formation.

Notably, many chaperones, as well as other proteins, can form foci in cells that have been grown to high density in YPD [23]. Many of these cases may be somewhat artifactual, similar to what we describe, or they may be proteins that are collected into small IPOD-like deposits; however, the amount of Ssa1 in these small foci may not be sufficiently high to allow detection under conditions of high cytoplasmic Ssa1 levels. 

The formation of large foci by Ssa1-GFP may also be useful as a model to study the machinery involved in the formation of large protein deposits, since the protein is very abundant and the foci very distinct, and we have characterized various conditions in which HADS form, do not form, and can be disassembled. 

## 4. Materials and Methods

### 4.1. Yeast Strains and Cultivation Conditions

Most of the experiments in this work used cells derived from the BY4741 strain (MATa his3-1 leu2-0 met15-0 ura3-0), with all of the –GFP fusion strains being taken from the Yeast GFP fusion collection [16]. Strains producing tagRFP-fusions were constructed from appropriate GFP-tagged strains by using a universal plasmid that replaces the GFP::HIS3 cassete from the Yeast GFP collection with a tagRFP::URA3 cassete, as described in [21]. The plasmids for expressing tagged VHL were a gift from D. Kaganovich; the plasmid for expressing tagged huntingtin was a gift from M. Sherman [24]. Centromeric plasmids expressing mutated Ssa1 (pRS316-SSA1-21 (L483W) [25]; pDJ169-SSA1-K69M [26,27]) were gifts from prof. Daniel Masison and prof. Daniel Jarosz.

The media used was YP (Yeast extract 1% (Dia-m, Moscow, Russia), Peptone 2% (Dia-m, Russia)), with various carbon sources (YPD–2% glucose (Dia-m, Russia), YP-Gly–2.5% glycerol (Dia-m, Russia)) and SC (Yeast nitrogen base–1.7 g/L (Difco, Detroit, MI, USA), ammonium sulfate–5 g/L (Dia-m, Russia), glucose–2% (Dia-m, Russia), with addition of appropriate amino acids). When necessary, solid medium was prepared by including 2% agar (American-type, Panreac, Spain). Strains for colocalization studies were obtained by switching the mating type of the obtained tagRFP-fusion strain with pGal-HO, a plasmid encoding the HO-endonuclease gene under the control of an inducible GAL1 promoter, making it possible to switch mating type and perform mating with appropriate GFP-fusion strains. Diploids were selected on SC–His,-Ura medium. To obtain haploids, the diploids were sporulated in liquid 1% potassium acetate (Dia-m, Russia) for 5 days. Then, after verifying the formation of asci, the diploids were suspended in distilled water mixed 1:1 with ether [28] to select against vegetative (non-spore) cells and incubated for 30 min. After collecting the aqueous phase, cell suspensions were plated onto SC medium lacking His and Ura to select for segregants encoding both fusion proteins, as verified by microscopy. The strains were confirmed to be haploid, as verified by assaying mating type.

Constructs encoding dominant-negative mutants of Ssa1 were introduced into the Hsp104-GFP strain from the Yeast GFP fusion collection [16], with an empty pRS316 vector as a control. The transformants of the BY4741 Hsp104-GFP strain with either pRS316, pRS316-SSA1-21 or pDJ169-SSA1-K69M plasmids (URA3 marker) were inoculated into 100 μL of SC-Glu medium (as biological triplicates), grown overnight and then diluted 20-fold and grown at 30 °C for either 2 h (log phase) or 12 h (stationary phase).

### 4.2. Microscopy

To visualize HADS, cells were spotted onto 2% agar pads [29] or other solid media, when noted specifically. Visualization was accomplished using a Zeiss AxioSkop 40 microscope, 100× oil immersion objective, NA = 1.2, using GFP and Texas Red filter sets.

To monitor HADS-like aggregation of Hsp104-GFP in the cells with Ssa1 dominant-negative mutants, the cell suspensions were imaged as 6–8 z-slices per each field of view, and maximum intensity projections were generated using FIJI software (ver. 1.53t). The number of cells with GFP foci and with all-diffuse GFP signal was counted using the FIJI Point Tool.

### 4.3. Electrophoretic Analysis of HADS Formation

Yeast cultures were grown in liquid medium, centrifuged and washed in water and lyzed by beating with glass beads in buffer A: 30 mM Tris-HCl, pH 7.4, 150 mM NaCl, 1 mM dithiothreitol and 1% Triton X-100. To prevent proteolytic degradation, the buffer was supplemented with 10 mM phenylmethylsulfonyl fluoride and Complete protease inhibitor cocktail (Roche Applied Science, Basel, Switzerland). Cell debris was removed by centrifugation at 1500× *g* for 4 min. After that, an aliquot of the lysate was kept as a total sample, while the remainder was centrifuged at 15,000× *g* for 20 min, separating the sample into the supernatant and pellet fractions. The pellet was washed with buffer A and spun again. SDS-PAGE was performed according to the standard protocol in 10% polyacrylamide gels. Coomassie staining was performed according to the colloidal Coomassie protocol [30].

## Figures and Tables

**Figure 1 ijms-24-12758-f001:**
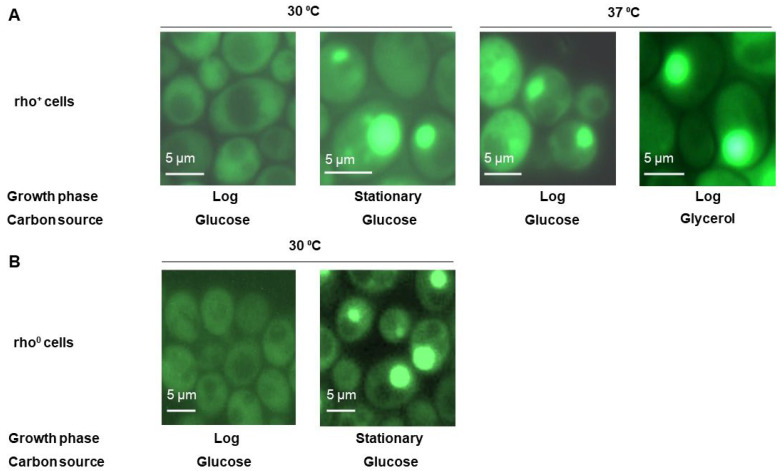
Ssa1-GFP is distributed diffusely in logarithmic YPD cultures and forms large foci under various mildly stressful conditions. (**A**) Cells of the BY4741 strain producing the Ssa1-GFP protein in YP medium containing the indicated carbon source were grown to the noted phase of growth (log – logarithmic or stationary) and visualized using fluorescence microscopy. (**B**) rho^0^ cells which lack their mitochondrial genomes due to passaging on ethidium bromide medium. Cells producing the Ssa1-GFP protein were grown in YPD medium to the noted phase of growth (log – logarithmic or stationary) and visualized using fluorescence microscopy.

**Figure 2 ijms-24-12758-f002:**
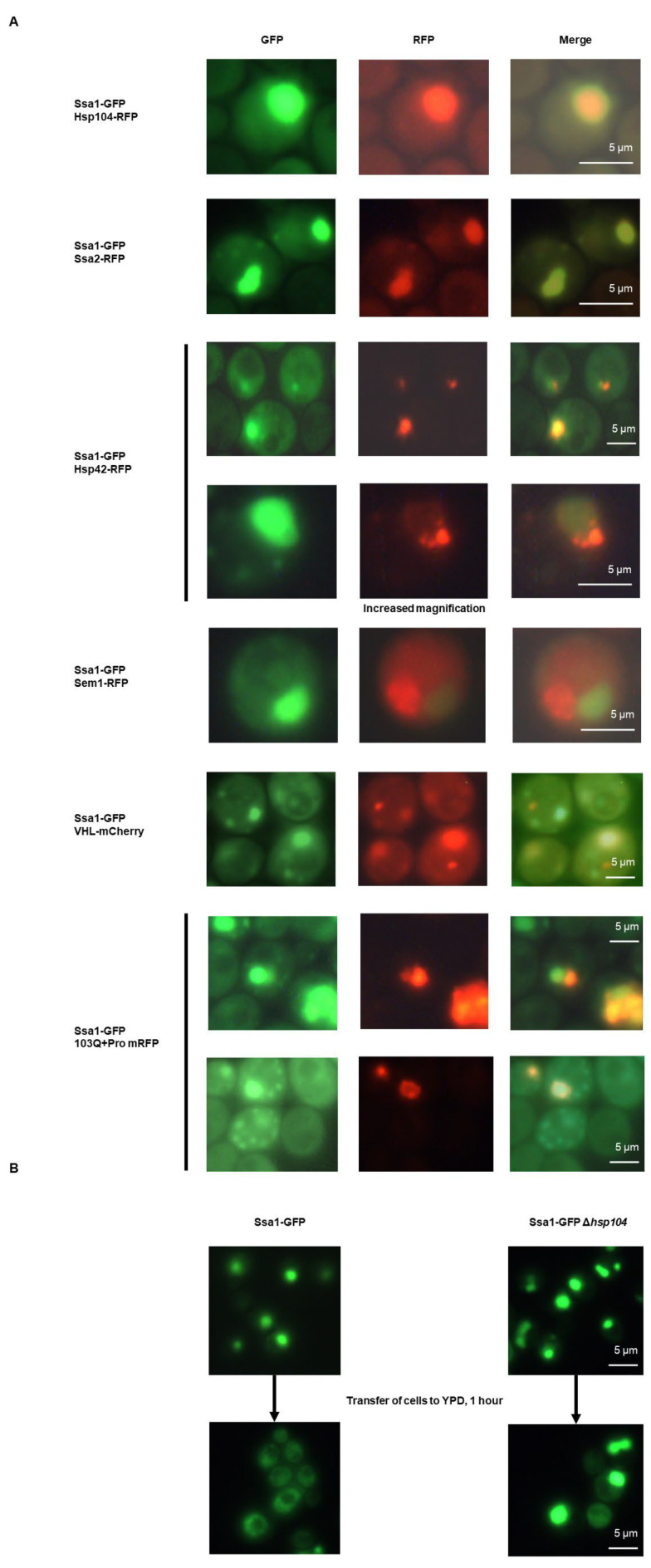
Ssa1-GFP HADS colocalize with various chaperones and misfolded proteins and are disassembled by Hsp104. (**A**) Cells of the BY4741 strain producing the indicated fusion proteins were grown in YP-Gly collected during the logarithmic growth phase and visualized using fluorescence microscopy. (**B**) Wild-type cells or cells with deleted *HSP104* expressing Ssa1-GFP were grown in YP-Gly medium up to the logarithmic phase and then either visualized immediately or after a 1 h incubation in YPD medium, which normally causes the dissolution of HADS.

**Figure 3 ijms-24-12758-f003:**
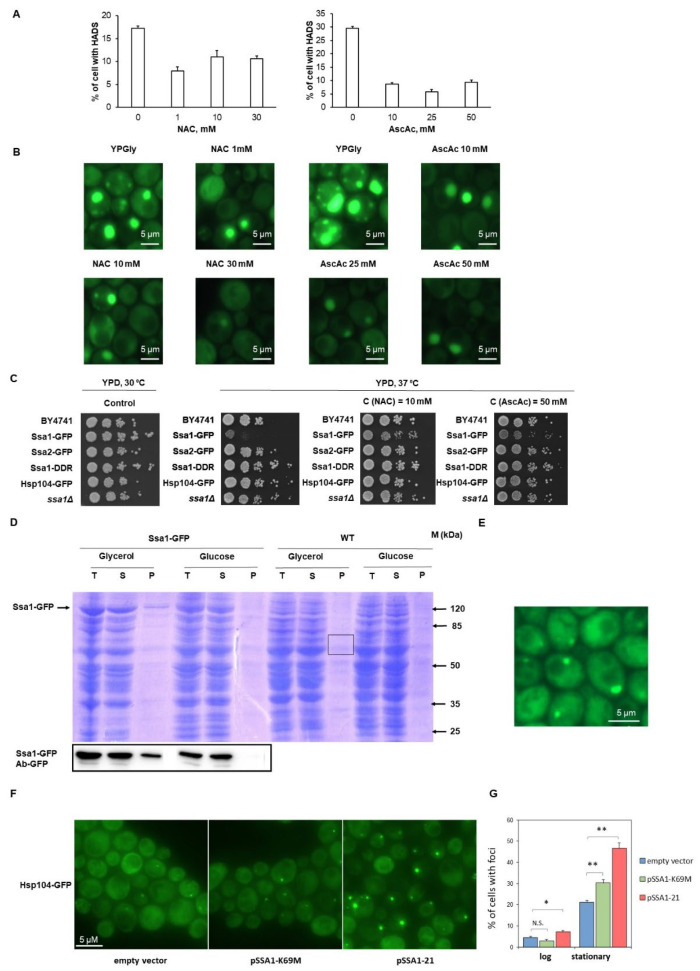
HADS formation can be mitigated by antioxidants and is likely a result of impaired Ssa1 function (**A**) Quantification of microscopy data from BY4741 cells expressing Ssa1-GFP, grown in YP-Gly medium, or the same with the addition of n-acetyl-cysteine (NAC) and ascorbic acid (AscAc). (**B**) Representative images from the results obtained in (**A**,**C**). (**C**) Serial five-fold dilutions of suspensions, containing BY4741 cells expressing the indicated fusion proteins, were incubated for 2 days on YPD with or without the addition of antioxidants at the indicated temperatures. (**D**) SDS-PAGE analysis of wild-type Ssa1 (WT) (right) and Ssa1-GFP in fractioned lysates obtained from wild-type and Ssa1-GFP expressing cells. Gel was stained with Coomassie blue or using the standard Western blotting procedure and anti-GFP antibodies for the visualization of Ssa1-GFP. (**E**) Cells of the BY4741 expressing the Ssa1-Dendra2 fusion protein were grown on solid YPD medium for 3 days, after which a sample of cells was visualized using fluorescence microscopy. (**F**) Stationary phase cells of the BY4741 strain producing Hsp104-GFP and the indicated Ssa1 mutant (or containing an empty vector). Cells were visualized using fluorescence microscopy; the representative maximum intensity projections obtained from a z-stack are shown. (**G**) Quantification of cells with Hsp104-GFP foci in either the logarithmic (log) or stationary growth phase. The experiment was performed in three biological replicates with 150–600 cells counted per replicate. Data are expressed as mean + SEM. * *p*-value < 0.05, ** *p*-value < 0.01, N.S.—non-significant (two-tailed Student’s *t*-test).

## Data Availability

Not applicable.

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
