# Peer review of "Fusion of Hsp70 to GFP Impairs Its Function and Causes Formation of Misfolded Protein Deposits under Mild Stress in Yeast"

_ijms, 2023, doi:10.3390/ijms241612758_

Round 1

Reviewer 1 Report

In this manuscript, Grosfeld EV et al. identified an insoluble protein deposit (IPOD)-like localization of Ssa1-GFP by using the Yeast GFP fusion library, where misfolded proteins aggregate in the steady state and they named it as HADS (Hsp70-associated deposit site). They examined the co-localization of Ssa1-GFP with proteasome and IPOD markers (Hsp42, Hsp104), and found that Ssa1-GFP did not localize with proteasome and partial localization with IPOD was observed.

While the results are interesting and suitable for publication, I could not understand the meaning of the sentence in lines 139 to 140 ("As noted above, . . ."). Please rewrite the sentence in a more understandable way.

Author Response

In this manuscript, Grosfeld EV et al. identified an insoluble protein deposit (IPOD)-like localization of Ssa1-GFP by using the Yeast GFP fusion library, where misfolded proteins aggregate in the steady state and they named it as HADS (Hsp70-associated deposit site). They examined the co-localization of Ssa1-GFP with proteasome and IPOD markers (Hsp42, Hsp104), and found that Ssa1-GFP did not localize with proteasome and partial localization with IPOD was observed.

While the results are interesting and suitable for publication, I could not understand the meaning of the sentence in lines 139 to 140 ("As noted above, . . ."). Please rewrite the sentence in a more understandable way.

We thank the reviewer for their comments. We have added some additional clarifications to the sentence, which now reads as – “As noted above, Ssa1-GFP foci colocalize with Ssa2-RFP (Figure 2) when these two proteins are coexpressed, however, to our surprise, on their own, Ssa2-RFP, nor Ssa2-GFP formed HADS either during growth on YP-Gly or in YPD at 37°C (data not shown), which are conducive to HADS formation by Ssa1-GFP”. We hope this makes things more clear.

In case the reviewer wants to view the changes made to the manuscript, we attach a file with tracked changes that include all of the changes to the manuscript that were made after the reviewer's comments. 

Reviewer 2 Report

This manuscript describes the characterization of the impact of addition of GFP to Ssa1, a member of the Hsp70 family.  The authors describe how addition of the GFP results in the construct forming insoluble deposits inside the cell, but hybrids developed with other proteins do not exhibit a similar behavior. They also report that Hsp104 can disrupt these deposits. Overall, these are interesting stories, and the science appears to be well-controlled.  However, the manuscript is lacking in a few key points.

First, there are several places where the authors state a conclusion, but it is not clear how that conclusion was reached.  Examples include: 

Lines 76-78 – the authors discuss assaying several types of proteotoxic stress but they don’t define what was tested.  I presume that temperature and glycerol are what the authors are referring to but I should not have to guess.  The authors should be more clear which variables they are referring to in the text.  Several also implies more than two, making me think there were more things tested than included in the manuscript.

Line 86 – It is not clear how absence of the mitochondrial genome might result in more oxidative species in the cell.

The description of the work in section 2.2 is very sparce and confusing.  There is a lot of data in Figure 2, which this section is based on, and there is a lot left up to the reader to interpret.  This paper would benefit from a more detailed description of the work and the conclusions.  Arrows highlighting the important features of the figures would also help.

Lines 121-122 – The authors mention that HADS assemble on YP-Gly but not on YPD, which I assume is from Figure 1, which is not stated. How do the authors know that the HADS don’t form at a lower level than on YPD, making them harder to detect.  The text reads like they don’t form at all but it is no clear how that conclusion was reached.

Line 142 – How was this conclusion reached?

Line 143-144 – How was it shown that large aggregates were not formed?

Lines 155-161 – How are the authors sure that aggregates were not formed that were not capable of being detected by centrifugation?

Overall, the work is interesting but the writing is extremely sparce, making the significance of the work less obvious based on the writeup as it currently stands.

There are a couple places where the language is awkward, such as Line 92. 

Author Response

This manuscript describes the characterization of the impact of addition of GFP to Ssa1, a member of the Hsp70 family.  The authors describe how addition of the GFP results in the construct forming insoluble deposits inside the cell, but hybrids developed with other proteins do not exhibit a similar behavior. They also report that Hsp104 can disrupt these deposits. Overall, these are interesting stories, and the science appears to be well-controlled.  However, the manuscript is lacking in a few key points.

We thank the reviewer for all of the comments and the favorable assessment of the paper.

First, there are several places where the authors state a conclusion, but it is not clear how that conclusion was reached.  Examples include: 

Lines 76-78 – the authors discuss assaying several types of proteotoxic stress but they don’t define what was tested.  I presume that temperature and glycerol are what the authors are referring to but I should not have to guess.  The authors should be more clear which variables they are referring to in the text.  Several also implies more than two, making me think there were more things tested than included in the manuscript.

We have rephrased the paragraph to be more clear. It now reads - Growth on non-fermentable carbon sources increases the amount of oxidatively damaged proteins [19]. This suggested to us, that other types of proteotoxic stress might also trigger HADS formation, and, indeed, we discovered that growth under heat-stress conditions (37°C) of log-phase cultures growing in YPD (glucose as the sole carbon source) also resulted in foci formation (Fig.1A).

Line 86 – It is not clear how absence of the mitochondrial genome might result in more oxidative species in the cell.

This data was obtained by the authors of the paper referenced in the text. In order to allow the reader to more easily understand this, we have added some of the details into the sentence. It now reads – “This might be due to the fact that absence of the mitochondrial genome, and the resulting mitochondrial dysfunction, increases the amount of reactive oxidative species in the cell, as evidenced by increased fluorescent of several redox-sensitive dyes as well as increased amounts of carbonylated proteins in rho0 cells (added reference 10.1016/j.cmet.2013.07.005 )  [20]. The oxidative stress was also shown to depend on RAS signaling and specifically Yno1, an ER-residing NADPH oxidase [20].”   

The description of the work in section 2.2 is very sparce and confusing.  There is a lot of data in Figure 2, which this section is based on, and there is a lot left up to the reader to interpret.  This paper would benefit from a more detailed description of the work and the conclusions.  Arrows highlighting the important features of the figures would also help.

We have added a small sentence to the introduction to highlight the difference between IPOD and JUNQ (one of which is the involvement of the proteasome as well as colocalization with proteasomal markers). We have also modified the phrasing of section 2.2 to make our logic a bit more clear. We thank the reviewer for helping us clarify this.

Lines 121-122 – The authors mention that HADS assemble on YP-Gly but not on YPD, which I assume is from Figure 1, which is not stated. How do the authors know that the HADS don’t form at a lower level than on YPD, making them harder to detect.  The text reads like they don’t form at all but it is no clear how that conclusion was reached.

A representation of this fact (no HADS formation of YPD in conditions of logarithmic growth) is presented in Figure 1 and, indeed, almost no cells had large aggregates of Ssa1-GFP in these conditions.  

Line 142 – How was this conclusion reached?

Reviewer #1 also commented on this line, so we modified the wording. It now reads - “As noted above, Ssa1-GFP foci colocalize with Ssa2-RFP (Figure 2) when these two proteins are coexpressed, however, to our surprise, on their own, Ssa2-RFP, nor Ssa2-GFP formed HADS either during growth on YP-Gly or in YPD at 37°C (data not shown), which are conducive to HADS formation by Ssa1-GFP”. We would also note that at this point in the story, it is not a conclusion, but rather a suggestion indicated by the data that is then confirmed by further experiments.

Line 143-144 – How was it shown that large aggregates were not formed?

In short, unlike Ssa1-GFP, Ssa1-Dendra does not form HADS in YP-Gly and that was observed in the referenced paper using fluorescent microscopy. Ssa1-Dendra expressing cells were extensively used in in that work performed at our lab. In that paper, Ssa1-Dendra was to study a phenomenon (formation of osmotic stress-induced foci) originally observed for Ssa1-GFP. That work was actually hampered by HADS formation, since most of the experiments needed to be conducted in YP-Gly, and Ssa1-Dendra proved to be a solution to this.

Lines 155-161 – How are the authors sure that aggregates were not formed that were not capable of being detected by centrifugation?

We agree with the reviewer that we cannot be sure of this, neither do we make such a claim in the text. A solid conclusion we can make is that WT Ssa1 does not form large aggregates in similar amounts and properties to the HADS formed by Ssa1-GFP. Possibly it can form smaller aggregates that do not precipitate as easily during centrifugation. We have added this clarification to the text. It reads – “This allows us to conclude that wild-type Ssa1 does not form comparable amounts of large aggregates that are similar to HADS. We cannot exclude the possibility of it forming smaller aggregates with distinct sedimentation/stability characteristics.” Figure 3F demonstrates that Ssa1 impairment causes formation of small aggregates labelled by Hsp104, so we do think that smaller aggregates of Ssa1 are highly likely. We have added a few sentences to this purpose as well. These read - The small size of the aggregates might also explain why we could not detect large amounts of sedimented wild-type Ssa1 in cells grown in YP-Gly (Figure 3D). The small size of the aggregates would reduce both the share of the protein involved in the aggregates, as well as, possibly, their stability during and after cell lysis.   

Overall, the work is interesting but the writing is extremely sparce, making the significance of the work less obvious based on the writeup as it currently stands.

We thank the reviewer for their efforts in helping us improve the clarity of the text, as well for the overall favorably view on our work.

Comments on the Quality of English Language

There are a couple places where the language is awkward, such as Line 92. 

We agree with the reviewer, we have modified the wording to be more clear.

We attach a file with the tracked changes of the manuscript so that the reviewer can look at the changes made to the manuscript based on the reviewer's comments.